# Adaptive Duration Modification of Speech using Masked Convolutional Networks and Open-Loop Time Warping

*Ravi Shankar[1], Archana Venkataraman[1]*

[1]Department of Electrical and Computer Engineering, Johns Hopkins University, Baltimore, MD

`rshanka3@jhu.edu, archana.venkataraman@jhu.edu`

## Abstract

We propose a new method to adaptively modify the rhythm of a given speech signal. We train a masked convolutional encoder-decoder network to generate this attention map via a stochastic version of the mean absolute error loss function. Our model also predicts the length of the target speech signal using the encoder embeddings, which determines the number of time steps for the decoding operation. During testing, we use the learned attention map as a proxy for the frame-wise similarity matrix between the given input speech and an unknown target speech signal. In an open-loop fashion, we compute a warping path for rhythm modification. Our experiments demonstrate that this adaptive framework achieves similar performance as the fully supervised dynamic time warping algorithm on both voice conversion and emotion conversion tasks. We also show that the modified speech utterances achieve high user quality ratings, thus highlighting the practical utility of our method.

## 1. Introduction

Human speech is a rich and varied mode of communication that encompasses both semantic information and the mood/intent of the speaker. The latter attribute is primarily conveyed by prosodic features, such as pitch, energy, and speaking rhythm. Many open problems in speech rely on a deeper understanding of and ability to manipulate these prosodic features. Consider voice conversion and emotion conversion systems. Pitch and energy modifications can be used to inject emotional cues into the neutral speech or to change the overall speaking style [1, 2, 3, 4, 5]. Prosodic features are also used to evaluate the quality of human machine dialog systems [6], and they play a significant role in speaker identification and recognition [7]. Rhythm, in particular, plays a crucial role in conveying emotions [8] and in diagnosing human speech pathologies [9].

While there are many approaches for automated pitch and energy modification [10, 11, 12, 13, 14], comparatively little progress has been made in changing the rhythm of a speech utterance. In fact, rhythm is difficult to manipulate because, unlike pitch or energy, there is no explicit coding for the relative duration of phonemes across the utterance. Rather, this information is implicitly defined and varies dramatically across speakers and utterances. As a result, rhythm modification methods either require considerable user supervision or they are geared towards aligning to known speech signals. Even prior work on quantifying the transitory behavior of rhythm [15] is limited and requires *a priori* alignment of the audio files.

Perhaps the earliest duration modification method is the time-domain pitch synchronous overlap and add (TD-PSOLA) algorithm [16]. TD-PSOLA modifies the pitch and duration of a speech signal by replicating and interpolating between individual frames centered at the peaks of auto-correlation signal. However, the user must manually specify both the portion of speech to modify and the exact manner in which it should be altered. Methods such as [17, 18] take a more user-friendly and performative approach to modify the pitch and rhythm, but they still require manual input to guide the process. An alternate approach to changing rhythm is a framewise alignment between a source utterance and a given target. Here, the most common approach is Dynamic Time Warping (DTW) [19]. It is a dynamic programming approach to align two sequences of different lengths. DTW requires both, the source and target speech which renders it unusable for generative modeling.

Finally, recent advancements in deep learning have led to a new generation of neural vocoders that disentangle the semantic content from the speaking style [20, 21, 22]. These vocoders can alter the speaking rate via the learned style embeddings. While these models represent seminal contributions to speech synthesis, the latent representations are learned in an unsupervised manner, which makes it difficult for the user to control the output speaking style. Another drawback is that these methods require large amounts of data and computational resources for adequate model training and speech generation [23, 24].

In this paper, we introduce an automated and adaptive speech duration modification scheme. Our approach combines the structured simplicity of dynamic decoding with the representation capabilities of deep neural networks. Namely, we model the alignment between a source and target utterance via a latent attention map; these maps are used as replacement of the similarity matrix for backtracking. We train a masked convolutional encoder-decoder network to estimate these attention maps using a stochastic mean absolute error (MAE) formulation. Unlike the conventional DTW [19] algorithm, once trained our framework operates in an open-loop fashion on the source utterance without needing access to the target. We demonstrate our framework on a voice conversion task using the CMU-Arctic dataset [25] and on three multi-speaker emotion conversion tasks using the VESUS dataset [26]. Our experiments confirm that the proposed model can perform adaptive duration modification with limited training data and minimal distortion.

## 2. Method

Our technique uses an attention based encoder-decoder framework to process an input sequence and produce another sequence as output. Specifically, the input sequences used in our model are the Mel-frequency representation of a speech signal. We further inject domain knowledge or prior into the neural network model by restricting the scope of the attention map between the encoded and decoded representations and strategically leverage DTW to generate intelligible speech. We provide a brief description of the training/testing strategy followed by a

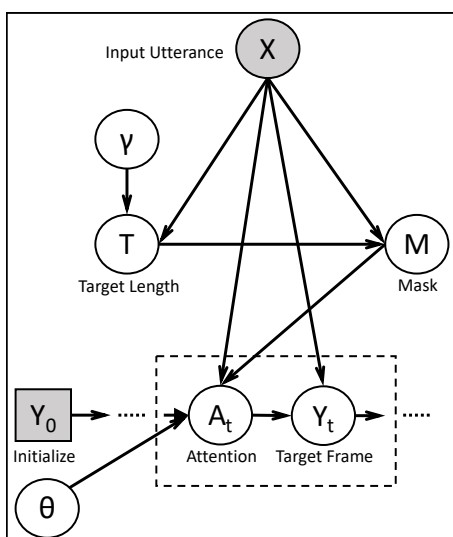

Figure 1: *Graphical model for rhythm modification. $\gamma$ and $\theta$ are the model parameters inferred during training. Attention $A_t$ is conditionally independent of target length $T$ given $X$ and $M$*

discussion of the baseline methods at the end of this section.

Fig. 1 illustrates our underlying generative process. Given an utterance $X$, we first estimate the length $T$ of the (unknown) target utterance $Y$ and subsequently use it to estimate a mask $M$ for the attention map. The mask restricts the domain of the attention vectors $A_t$ at each frame $t$ during the inference stage to mitigate distortion. We use paired data $(X_{tr}, Y_{tr})$ to train a convolutional encoder-decoder network to generate the attention vectors. During testing, we first generate the attention map from the input $X$ and use it to produce the target speech $Y$.

### 2.1. Loss Function

Formally, let $X \in \mathbb{R}^{D \times T_s}$ denote the frame-wise Mel filter-bank energies extracted from the input speech. Here, $D$ is the number of filter banks, and $T_s$ is the number of temporal frames in the utterance. Similarly, we denote the target speech as $Y \in \mathbb{R}^{D \times T}$, where the target length $T$ is usually different from $T_s$.

Our generative process for the target speech is as follows:

$$T \sim \text{Laplace}(T^0, b_T) \quad and \quad Y_t \sim \text{Laplace}(Y_t^0, b_y) \ \forall t, \ (1)$$

where $Y_t$ is the target Mel filter-bank energy features at time $t$. We use Laplace distributions to leverage sparse nature of filter-bank energies. The parameters $\{T^0, b_T, Y_t^0, b_y\}$ of the distributions are unknown and implicitly estimated via a deep neural network, which is parameterized by $\gamma$ and $\theta$ (see Fig. 1).

By treating the unknown parameters as functions of the input $X$, we obtain the following estimating equations for the target sequence length and frame-wise Mel filter-bank energies:

$$\hat{T} = f_\gamma(X) \quad and \quad \hat{Y}_t = X \cdot A_t + f_\theta(X, \hat{Y}_{0:t-1}). \ (2)$$

The functions $f_\gamma(\cdot)$ and $f_\theta(\cdot, \cdot)$ correspond to the length prediction and energy estimation component of the neural network. The variable $A_t \in \mathbb{R}^{T_s}$ is an attention vector that combines frame-wise features of the source utterance $X$ to generate the target frame $\hat{Y}_t$. Our model differs from standard sequence-to-sequence model by treating the neural network predictions as residuals added to input sequence itself, where these residuals

depend on input and the history of predictions $\hat{Y}_{0:t-1}$. This autoregressive property allows the neural network to learn both segmental and supra-segmental variations that can potentially distinguish between different speakers or emotions.

During training, we use paired data $(X, Y)$ and maximize the likelihood of the target speech signal with respect to the neural network weights $\{\theta, \gamma\}$. This likelihood can be written

$$P(\hat{Y}, \hat{T}|X) = P(\hat{T}|X) \prod_{t=1}^{\hat{T}} P(\hat{Y}_t|X, \hat{T}, \hat{Y}_{0:t-1}), \quad (3)$$

where, the second term in Eq. (3) can be obtained by introducing a deterministic attention mask $M$ and marginalizing $A_t$:

$$P(\hat{Y}_t|X, \hat{T}, \hat{Y}_{0:t-1}) = \sum_{A_t} P(\hat{Y}_t, A_t|X, \hat{T}, \hat{Y}_{0:t-1}, M)$$
$$= \sum_{A_t} P(\hat{Y}_t|X, \hat{T}, A_t, \hat{Y}_{0:t-1}) P(A_t|X, \hat{Y}_{0:t-1}, M) \quad (4)$$

The variable $M$ here denotes the attention mask. We introduce $M$ for mathematical convenience, as it is a deterministic function of the source length $T_s$ and the estimated length $\hat{T}$. We encode the attention $A_t$ as a one-hot vector across the $T_s$ frames of the source speech. Thus, it follows a categorical distribution. For simplicity, we model $A_t$ as conditionally independent of the target length $\hat{T}$ given the mask $M$ and the input $X$. Taking the $\log(\cdot)$ of likelihood term and combining with Eq. (4) yields:

$$\mathcal{L}(\theta, \gamma) = -\log\Big(\sum_{A_t} P(\hat{Y}_t, A_t|X, \hat{T}, \hat{Y}_{0:t-1}, M)\Big) - \log\big(P(\hat{T}|X)\big)$$
$$= -\log\Big(\sum_{A_t} \frac{q_\theta(A_t|X, \hat{Y}_{0:t-1}, M)}{q_\theta(A_t|X, \hat{Y}_{0:t-1}, M)} P(\hat{Y}_t, A_t|X, \hat{T}, \hat{Y}_{0:t-1}, M)\Big)$$
$$- \log\big(P(\hat{T}|X)\big)$$
$$\leq -\sum_{A_t} q_\theta(A_t|X, \hat{Y}_{0:t-1}, M) \log\big(P(\hat{Y}_t|X, A_t, \hat{Y}_{0:t-1})\big)$$
$$- \log\big(P(\hat{T}|X)\big) + KL(q_\theta(A_t)||P(A_t))$$
$$= -\sum_{A_t} q_\theta(A_t|X, \hat{Y}_{0:t-1}, M) \log\big(P(\hat{Y}_t|X, A_t, \hat{Y}_{0:t-1})\big)$$
$$- \log\big(P(\hat{T}|X) - H(q_\theta) + const.$$
$$\leq -\sum_{A_t} q_\theta(A_t|X, \hat{Y}_{0:t-1}, M) \log\big(P(\hat{Y}_t|X, A_t, \hat{Y}_{0:t-1})\big)$$
$$- \log\big(P(\hat{T}|X) + const.$$
$$(5)$$

The distribution $q_\theta(\cdot)$ above is an approximating distribution for the attention vectors implemented by a convolutional network. The first inequality uses the convexity of the $-\log$ function, and the second inequality comes from the fact that entropy $H(q_\theta) \geq 0$. Notice that we have implicitly assumed $P(A_t|X, \hat{Y}_{0:t-1}, M)$ has a uniform distribution over the masked region as a non-informative prior. This is a reasonable assumption given that the masking process reduces the attention domain to a small region. However, $q_\theta$ is **not penalized** for deviating from this uniform distribution prior during training. This flexibility allows the network to learn realistic attention vectors during autoregressive decoding. Eq. (5) can be easily translated into a neural network loss function which we minimize for $\{\theta, \gamma\}$:

$$\mathcal{L}(\theta, \gamma) = \lambda_1 \times E_{A_t \sim q_\theta}\big[\log\big(P(\hat{Y}_t|X, A_t, \hat{Y}_{0:t-1})\big)\big]$$
$$+ \lambda_2 \times \log\big(P(\hat{T}|X)\big)$$
$$= \lambda_1 \times E_{A_t}\big[\|\hat{Y}_t - Y_t^0\|_1\big] + \lambda_2 \times \|\hat{T} - T^0\|_1 \quad (6)$$

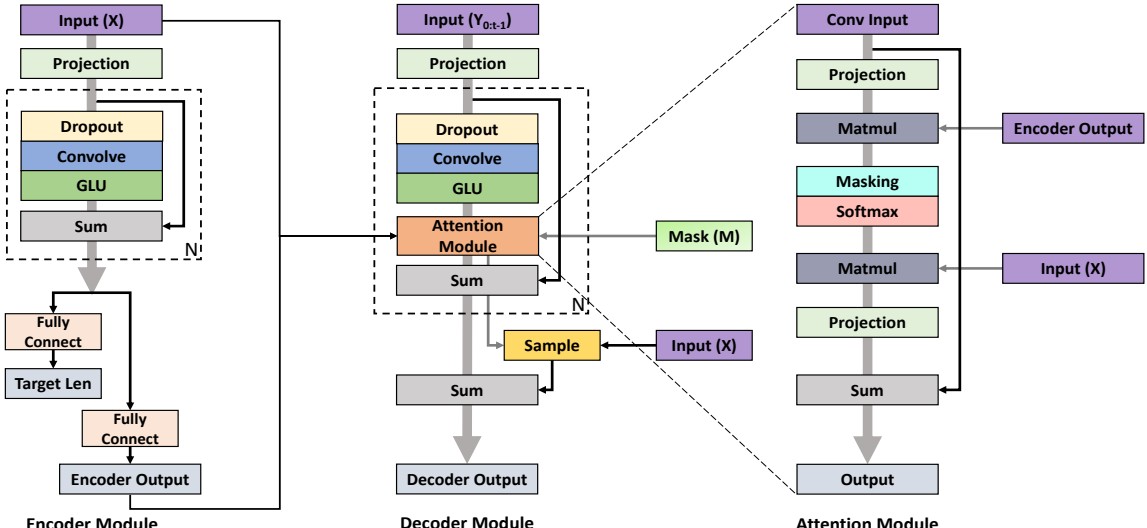

**Encoder Module**      **Decoder Module**      **Attention Module**

Figure 2: *Model architecture used for the sequence-to-sequence speech generation. The encoder and decoder modules consist of 10 identical blocks. Projection layers are simple feed-forward layers without any non-linearity to project input features in high dimension.*

where $\lambda_1$ and $\lambda_2$ are the model hyperparameters that adjusts the trade-off between the two objectives and contains the variances of the Laplace distributions. Notice that the loss in Eq. (6) computes an expectation over the attention maps. We use the Monte-Carlo estimate by sampling from the attention map at each time-step. The training procedure is therefore stochastic in nature due to this random sampling from the attention map.

### 2.2. Convolutional sequence-to-sequence model

We use a masked convolutional sequence-to-sequence model to learn the duration transformation from one domain to another. Fig. 2 shows the interplay between the encoder, decoder and modified attention modules of our deep neural network. The architecture is adapted from [27] by adding residual connections to the final layer and reconfiguring the attention module. The encoder in Fig. 2 is a stack of gated convolutions which performs two tasks: (i) approximating the length of the target sequence and (ii) learning appropriate representation for the decoding process. We insert an attention module between the encoder and decoder layers to leverage locality constraint during the generation of the output sequence. Notice that, a general attention map uses the entire input sequence to decode a single frame of the output sequence. Therefore, we apply a masking strategy inspired by Itakura parallelogram of DTW framework which acts as a prior knowledge over the difference in speaking rates between the source and target domains. Masked convolutions were initially proposed by [27] for language modeling. This architecture allows the network to exploit local continuity of speech and can be trained faster than a conventional RNN or LSTM while also requiring fewer learnable parameters. These advantages are important in cases of limited training data.

### 2.3. Masking

We use the mask $M$ to constrain the scope of the attention mechanism to be similar in time-scale to the input. This procedure is important for two reasons. From a speech quality perspective, large local swings in speaking rate may generate unintelligible speech. From an estimation perspective, the speech utterances contains hundreds (sometimes thousands) of frames.

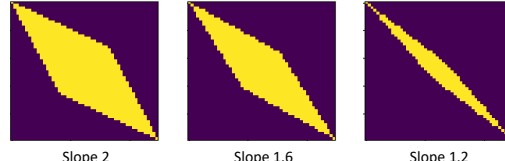

Figure 3: *Binary attention masks with 3 different slopes.*

It is difficult to robustly train a deep network to generate such long attention vectors based on smaller datasets.

These masks are derived from Itakura parallelogram [28], as illustrated in Fig. 3 and is different from [29] due to hard cut-off in scope. The slope of the Itakura parallelogram specifies the minimum and maximum speaking rates that the reconstructed utterances are allowed to possess in comparison to the input speech. In this paper, we fixed the minimum and maximum variation in speaking rate to 0.8 and 1.25, respectively, based on empirical observations of the training data.

### 2.4. DTW Back-Tracking

Our final step uses the learned attention map as a proxy for the DTW similarity matrix. This strategy allows us to train the model on a relatively small dataset (e.g., 2-3 hours) and still generate intelligible speech during open-loop modification of new utterances. Formally, we apply a dynamic programming operation to the attention maps produced by the neural network to get a path of alignment from source to target. To avoid skipping phonemes, we constrain the dynamic programming path to take at most one horizontal or vertical step at a time while back-tracking. Once estimated, the path informs a reorganization of the source utterance frames via localized contraction and dilation operations. Following this reorganization, the target speech is synthesized via the WORLD vocoder [30].

We train our model using mini-batch gradient descent and the Adam optimizer [31] with a fixed learning rate of $10^{-4}$ and a batch size of 16. The input $X$ are 80-dimensional Mel-filterbank energies spanning 0-8 kHz. The projection layer expands this input to 256 dimensions. Both the encoder and decoder consist of 10 convolutional layers, each followed by a

gated linear unit. Given the small dataset size, we use data augmentation to mitigate over-fitting. Specifically, we reverse the input-output sequences and randomly extract intervals of variable size (with probability 0.5) from the full speech utterance.

---

**Algorithm 1:** Strategy for model training

1  function trainModelParameters $(X, Y)$;
   **Input**  : filterbank energies ($X \in \mathbb{R}^{D \times T_s}, Y \in \mathbb{R}^{D \times T_t}$)
   **Output:** model parameters $(\theta, \gamma)$
2  **if** *epoch < MaxEpochs* **then**
3     **for** *minibatch* **do**
4        Predict target length $\hat{T} = f_\gamma(X)$ and create the mask $M \in \mathbb{R}^{T_s \times T_t}$;
5        Estimate $A \in \mathbb{R}^{T_s \times T_t}$ using masked convolution and sample $u \sim U(0,1)$;
6        **if** *u < 0.2* **then**
7           Sample $a \in \mathbb{R}^{T_s}$ from $A_{T_s}$;
8           Reconstruct using:
           $\hat{Y}_t = X \cdot a + f_\theta(X, Y_{0:t-1})$;
9        **else**
10          Reconstruct using:
           $\hat{Y}_t = X \cdot A_{T_s} + f_\theta(X, Y_{0:t-1})$;
11       **end**
12       Compute prediction errors and update parameters;
13    **end**
14    epoch ← epoch + 1;
15 **end**
16 return trainedModel;

---

### 2.5. Training and Testing Strategy

During training, we optimize Eq. 6 based on the Mel filterbank energies $Y$ and utterance durations $T$ from paired input-output utterances. The forward pass through the network (Fig. 2) processes the input frames and generates an embedding to predict the target sequence length $\hat{T}$. The embedding is also used to generate an attention vector as a categorical distribution at each decoder step inside the specified masked region. We use a stochastic sampling procedure for the attention vector, in which we randomly mix between a single sample from the distribution $q_\theta$ and the MAP estimate. Empirically, this strategy provides robustness to sub-optimal local minima (see Alg. 1).

During testing, we rely on the predicted length to generate the attention map and the target frames. We also use a MAP strategy, rather than the stochastic mixing procedure. Once generated, we use the attention map as a proxy for the DTW similarity matrix; using a Viterbi alignment procedure, we rearrange the input frames to produce the modified speech (Alg.2).

### 2.6. Baseline Comparison Methods

We compare our convolutional encoder with two commonly used sequence-to-sequence frameworks: (i) Gated Recurrent Unit or GRU model [32], and (ii) Transformer model [33]. Due to space limitations, further details of the baseline architectures and training strategy have been omitted from the paper.

---

**Algorithm 2:** Strategy for model testing (i.e., open-loop duration modification)

1  function modifyDuration $(X)$;
   **Input**  : filter-bank energy ($X \in \mathbb{R}^{D \times T_s}$ and $Y_0$)
   **Output:** alignments $((x_1, y_1), (x_2, y_2), ...)$
2  Predict length of target sequence $\hat{T}_t = f_\gamma(X)$;
3  Create attention mask $M \in \mathbb{R}^{T_s \times \hat{T}_t}$ and Set $t = 0$;
4  **if** $t < \hat{T}_t$ **then**
5     Using mask $M_t$, $X$, and $Y_{0:t-1}$ estimate $A_t$;
6     Using $X$, $Y_{0:t-1}$, and $A_t$, predict $Y_t$;
7     $t \leftarrow t + 1$;
8  **end**
9  Run DTW backtracking on the attention matrix $A$;
10 return (alignments $(x_1, y_1), (x_2, y_2), ...(x_n, y_n)$);

---

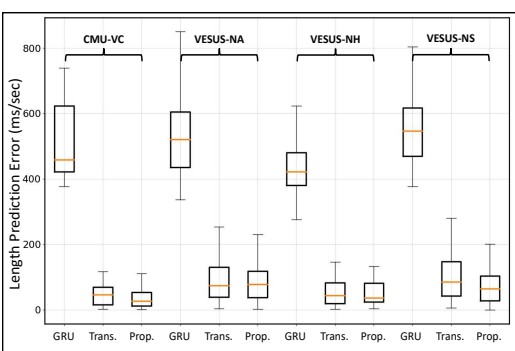

Figure 4: *Length prediction errors ($\downarrow$) across different models.*

## 3. Experimental Results

We evaluate our rhythm modification framework on two publicly available multi-speaker datasets: CMU-ARCTIC [25] for voice morphing and VESUS [26] for emotion conversion.

### 3.1. Data and Conversion Tasks

The CMU-ARCTIC database has 4 American English speakers (two male, two female), who we paired by gender for voice conversion. Of the resulting 2264 sentence pairs, we train our model and the baselines using 2164 utterances and reserve the remaining 100 utterances (random 50-50 split) for validation and testing of the open-loop modification properties.

VESUS is an emotional speech corpus containing utterances in 4 emotion classes: neutral, angry, happy, and sad. Each utterance contains 10 crowd-sourced ratings of emotional saliency For robustness, we only use utterances that are correctly annotated by at least half of the listeners. We consider three neutral-emotional conversion tasks as follows:

- **Neutral to Angry**: 2385 utterances for training, 72 for validation and, 61 for testing.
- **Neutral to Happy**: 2431 utterances for training, 43 for validation and, 43 for testing.
- **Neutral to Sad**: 2371 utterances for training, 75 for validation and, 63 for testing.

Due to the smaller sample size, we pretrain the models on CMU-ARCTIC and fine-tune it for emotion conversion.

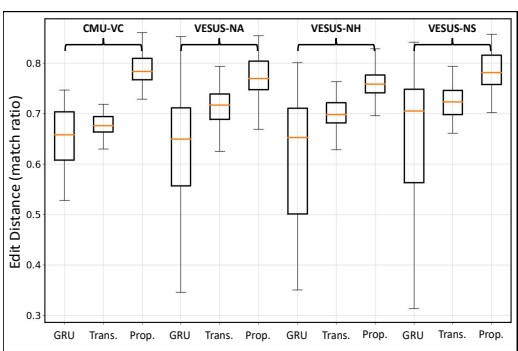

Figure 5: *Alignment similarity* (↑) *between attention and DTW.*

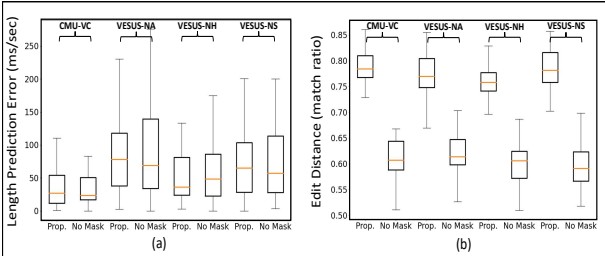

Figure 6: *(a) Length prediction of target utterances* (↓) *and (b) measuring similarity of attention map* (↑) *to DTW cost matrix. Model is trained without mask constraint on attention map.*

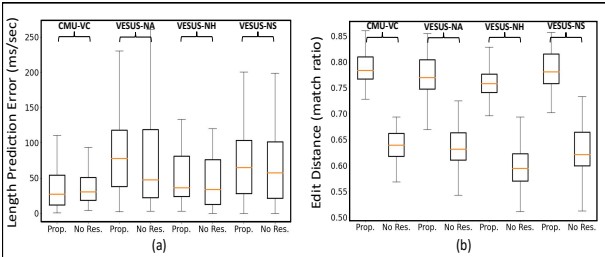

Figure 7: *(a) Length prediction of target utterances* (↓) *and (b) measuring similarity of attention map* (↑) *to DTW cost matrix. Model is trained without residual connection in decoder layer.*

### 3.2. Length Prediction

As a sanity check, we compare the predicted utterance length by our framework with that of the ground truth parallel utterance. Fig. 4 shows the error in predicting the length ratio in a ms/sec format. Notice that, our framework mispredicts the utterance lengths by only 40ms/sec and 65ms/sec (on average) on CMU-ARCTIC and VESUS, respectively. Duration prediction is particularly challenging on VESUS due to marked differences between neutral and emotional utterances. The median prediction error for GRU model is in the range of $400-600$ms per second of the input utterance. The Transformer fares relatively well in comparison to GRU because of its ability to establish long-range dependency. However, our framework performs slightly better, perhaps due to the multi-task setup and the fusion of deep representation with Bayesian regularization.

### 3.3. Attention Alignment

Next, we compare the open-loop alignment estimated via the attention map with the supervised DTW algorithm where both utterances are known. To compare the warping paths, we code the horizontal, diagonal, and vertical moves of the backtracking procedure into three classes. We then compute the edit distance between the attention map and DTW-based alignment schemes. Fig. 5 illustrates the match ratio normalized by the average length of sequences. As seen, the match ratio varies between 0.70 and 0.85, which suggests that our convolutional model can readily learn the general characteristics of duration modification. The GRU model performs poorly in this task due to its inability to learn sequence transformations across 100s of frames. The Transformer model does a little better than the GRU on this task, but still underperforms our method, likely due to the small training dataset. Our proposed model performs best because of the Itakura masking constraint and its reduced parameterization, which permits learning in small-data regimes. Thus, our method can be used as a tool for manipulation of speaking rate at both, local and global scale.

### 3.4. Ablation Analysis: Removing Itakura masking

There are multiple components in the proposed model which work in synchronised manner to produce naturally sounding speech. In addition to the generative modeling, the two most important augmentations we have made to the masked convolutional network pipeline are: (i) using Itakura masking for attention map and (ii) using an attention weighted residual connection in the final layer. Therefore, we perform ablation experiments to understand the relative significance of each of these

augmentations. Our first experiment removes masking from the attention layers. Fig. 6 shows the model's performance on target length prediction and approximating the DTW similarity matrix. The results in Fig. 6(a) indicate that the length prediction performance is roughly similar to the proposed model. This is expected because, the encoder part of the network is exactly same. The attention map is constrained only in the encoder-to-decoder transition. Hence, it estimates length with a relatively small error (in ms/sec). The match ratio metric however, (shown in Fig. 6(b)) is considerably worse. Itakura masking procedure acts as a good inductive bias/prior on the attention map because the speech rate do not fluctuate drastically in human conversations. Therefore, our localization scheme for the attention map is crucial to improve the edit distance in our model.

### 3.5. Ablation Analysis: Removing Residual connection

Our second ablation experiment involves removing the attention weighted residual connection from the final layer of decoder. Fig. 7(a) shows that the model is able to estimate the target sequence lengths with a relatively low error rate. We attribute this to the fact that the encoder portion is same as proposed model. The match ratio (Fig. 7(b)) in this experiment is slightly better than the no-masking results but, worse than the proposed model. Therefore, we can confidently say both Itakura masking scheme and residual connection helps in approximating DTW similarity matrix. Further, the presence of residual connection is extremely useful in providing a good gradient signal for the convolutional network to learn prediction of target frames. Since the linguistic content of input and target utterances are same, it further allows the neural network to inherit input speech properties which is helpful in auto-regressive generation mode.

### 3.6. Component-Wise Duration Analysis

Fig. 8 compares the differences in duration between the converted utterances and the ground truth targets for vowels, con-

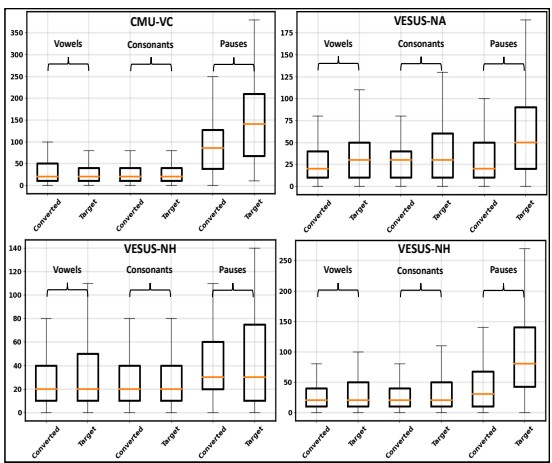

Figure 8: *Duration differences between source/target and source/converted pairs for vowels, consonants, and pauses.*

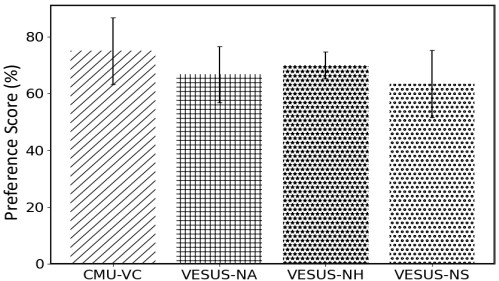

Figure 9: *Preference score (in %) of proposed method (↑) relative to the input with ground-truth as reference (crowd-sourced).*

sonants and short pauses. We use the Penn Phonetic Forced Alignment tool [34] to get the text and speech alignment. As seen in Fig. 8, our method faithfully modifies the duration of vowels and consonants, but it is less effective with short pauses. This trend is intuitive, as our model relies on replication of the frames determined by the backtracking on similarity map. Therefore, it cannot create pauses if these frames do not exist in the source utterances. Nonetheless, our model consistently estimates the difference between vowels and consonants duration across multiple tasks, which corroborates our claim of developing a general purpose speech rate manipulation framework.

### 3.7. Rhythm Similarity Assessment

To evaluate the rhythm of modified speech, we design a crowd-sourcing based preference test scheme. In this experiment, the evaluators are asked to listen the ground-truth speech as a reference. It is then followed by a selection task between the unmodified (source) and the modified utterance, whichever has the highest perceptual similarity to the reference in terms of speaking rate modulation. The results of this experiment are demonstrated in Fig. 9. We can note that, the similarity scores are in the range of 60-80% which is relatively high, considering that source and ground-truth utterances have duration difference in the order of 100-200ms only. CMU-VC task has the highest similarity score mainly because of the long utterances that allow the listeners to discern the differences in an effective manner.

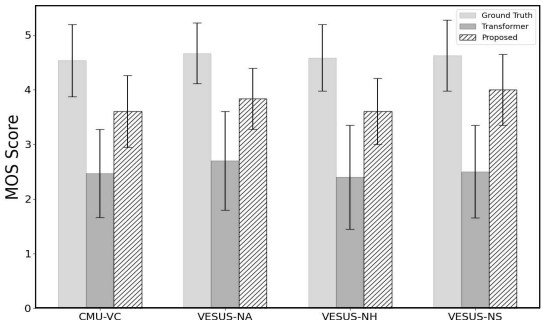

Figure 10: *Crowd-sourced MOS (↑) of generated speech (hatched bars) vs the ground-truth samples from each task (shaded left) and baseline transformer model (shaded middle).*

### 3.8. Speech Reconstruction Quality

Finally, we use crowd sourcing to obtain a mean opinion score (MOS) for the re-synthesized speech quality of the testing utterances. The crowd sourcing was performed using Amazon mechanical turk (AMT). We collect 5 listener ratings for each converted utterance in the test set, and we also add clean (ground-truth) along with some noisy/distorted utterances to the converted samples set to get the baseline scores and flag non-invested listeners and bots on AMT. As seen in Fig. 10, our method achieves an average MOS between $3.7 - 4.0$ across the four tasks (rightmost bars). Further, the ground-truth baseline score of each task (leftmost bars) are in the range of $4.5 - 5$, whereas the MOS score of speech generated by transformer model (middle bars) are in the range of $2 - 3$. It shows the superiority of proposed model over transformer baseline. We note that CMU-ARCTIC task has the lowest MOS, possibly due to longer and more complex utterances. Interestingly, the MOS is unaffected by errors in length prediction, as evidenced by the VESUS neutral-angry emotion conversion task. Thus, our model provides a robust way to alter speech characteristics.

## 4. Conclusions

We have introduced a new framework for adaptive rhythm modification. Our model used an attention based convolutional encoder-decoder architecture to estimate attention maps which associate frames of the input speech with frames of the target speech. The attention maps are modeled as latent variables in a graphical framework, which lead to a stochastic formulation of the mean absolute error (MAE) loss for model training. During testing, the attention map is directly used as an approximation of the similarity matrix for a DTW-style backtracking procedure. We evaluated our framework on a voice conversion and three separate emotion conversion tasks using CMU-ARCTIC and VESUS corpora. Our evaluation metrics are the L1 distance for target length prediction, and an edit distance based matching ratio for path similarity. Our proposed model outperformed existing seq-2-seq models designed solely on transformer and LSTM architectures in both metrics. Further, we ablate our proposed model's performance against simpler versions of it, i.e., no residual connection and no Itakura masking scheme. These ablations showed that removing either of these components leads to poor match ratio performance. Overall, our framework produced similar duration modification as the vanilla DTW, but *without requiring access to the target utterance*. Finally, we showed that the re-synthesized speech had similar naturalness to most state-of-the-art neural vocoders.

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
