# OpenReview forum: "Adaptive Duration Modification of Speech using Masked Convolutional Networks and Open-Loop Time Warping"
_Interspeech.org/2023/Workshop/SSW — SSW12_

### Official Review · Reviewer_Pyry · 2023-05-30
**Method for rhythm modification based on an encoder-decoder architecture with masked attention**

**Rating:** 5
**Confidence:** 4

**Review:**

The paper proposes an encoder-decoder architecture to modify the rhythm of input spectrograms by applying a masked attention with controlled slope. The system is able to predict the duration of the modified sentence. With this predicted duration it also estimates the attention map that is used in a backtracking procedure to calculate the alignments that will be used to reorganize the input frames obtaining a new version with a modified rhythm.

The method seems sound, and good results are obtained. My main concern is related to the subjective evaluation. In addition, there are some additional aspects that need clarification:

-	Why has the WORLD vocoder been used? Why not use a pre-trained (and maybe fine-tuned in a reduced amount of data) neural vocoder used instead? Neural vocoders produce signals with better quality than that provided by previous vocoders as WORLD. Big data amounts are needed to train them from scratch, but not to fine tune them.
-	What do you mean by restricting the scope of the attention map in the first paragraph of section 2? Do you mean slope?
-	In algorithm 2, what is the input Y_0? Where does it come from? And why this function has no parameter to control the amount of duration modification as input? If the modifications are truly controllable, as it is stated in the paper, some way of controlling it must be provided

I have some reservations about the MOS evaluation:

-	The evaluators were asked to rate the speech quality of the signals. They were not told to take special care evaluating the rhythm of the samples. Thus, the results provide no clue about whether the evaluators have considered the generated signals have proper rhythm variations. If for the evaluators differences in rhythm are not important when evaluating quality, the result of this evaluation is not addressing the changes produced by the proposed method. A proper question is critical when designing a MOS evaluation.
-	Comparing the results of this evaluations with MOS scores obtained in other evaluation campaigns (as it is done to claim that this work obtains results comparable to those of neural vocoders) is not a good practice. The evaluation must contain signals from all the methods that want to be compared and the results must be tested for statistical significance to be able to extract valid conclusion.
-	An even bigger flaw is the absence of natural sentences (even if recoded) or at least sentences modified using the actual DTW alignment to set up a ground truth baseline. This is very likely the cause of the good MOS scores as without a reference of actual good quality it is very difficult for evaluators to calibrate their scores. Besides, including these good quality signals in the evaluations helps filtering out non-invested evaluators and could help locating these evaluators in addition to or instead of the noisy signals used in the evaluation.

MINOR COMMENTS

References [6], [15], [17], [23] and [26] lack details about the journal or conference (or at least arXiv link or similar) where they have been published.

There is a sentence in the conclusion section that references to supplementary material, but there is no supplementary material provided with the paper.

---

### Official Review · Reviewer_Eu9e · 2023-06-04
**Interesting but lacking in some respects**

**Rating:** 5
**Confidence:** 4

**Review:**

The architecture proposed for adaptive rhythm modification is based on existing masked convolutional encoder-decoder network (reference [26] of the paper). The authors propose to add a few modifications:
1-  Use of a masking constraint to force the attention into a near diagonal region.
2 - Add a residual connection in the final layer
3 - At training, the attention map is treated as a distribution and stochastic sampling is used to increase robustness, the sampled path defines a warping between the input and output sequences (mels spectrograms). At inference, dynamic programming is used to find the best path without skips from the attention map and perform the warping.

Originality:
In terms of novelty, the first point is not really new as such constraint has been proposed in attention-based sequence-to-sequence TTS for instance. The most interesting aspect in the point 3 but it could be argued that it introduces a gap between training and inference. Techniques using CTC-based layers such as in the "One TTS Alignment To Rule Them All" could be employed instead here. It should be noted that this paper is not cited despite being quite relevant for the proposed approach. (More generally, the paper lacks recent references, most of the references being 3 years old or more).

Clarity: The paper is globally well written (apart for some confusing terms, like using the term "vocoder" to refer to Tacotron for instance). However, some aspects of the architecture lack clarity. Especially in the decoder, when it comes to the sampling procedure. The paper provides an algorithmic description of the strategies used at  training and inference but some notations are not introduced which makes it hard to fully comprehend. And the diagram on Figure 1 is not helping on this respect as it doesn't show the difference between training and inference.

Quality:
The proposed method seems technically correct and the experimental part is thorough but lacking in some key points.
First, contrary to what the authors write, the results on length prediction do not show significant improvement between the proposed method and the transformer-based one. The proposed method outperforms the baseline approaches on alignment similarity between the attention map and DTW, but that's expected by design. The main problem is the subjective test that only evaluates the proposed method and doesn't compare it to any baseline like the Transformer-based method. In practice the MOS scores are always dependent of the systems being evaluated at the same time by the listeners, as they will adjust the range of the ratings accordingly. So here it's not really possible to compare between systems that weren't evaluated in the same test.
Finally, in addition to the CMU-ARCTIC dataset, the authors evaluate the proposed approach on emotional conversion task (neutral to happy, neutral to sad,  neutral to sad), but they should mention that the modification of rhythm is by no means enough to make such conversions.

---

> ### Author Response · Authors · 2023-06-23
> **Clarifications on technical contributions and extra experiments**
>
> We thank the reviewer for giving their feedback in a comprehensive manner. To highlight our technical contributions, we have added a full page of generative modeling of the modification process along with extra details and derivation of the loss function in the methods section. We omitted the details in submitted draft due to space constraint. This will also resolve notation issues in the proposed algorithms. Further, we have added a reference to the paper titled "One TTS Alignment to Rule Them All" which is different from our masking strategy. First, our masking strategy stems from the perspective of constraint on the minimum/maximum speaking rate in modified speech, which leads to Itakura parallelogram structure. Unlike the mentioned paper, our objective is not about keeping the alignment as close to the diagonal as possible rather, it to make sure that the alignment stays within the masked region as a whole. Second, the mask is a deterministic variable conditioned upon source and estimated target length unlike the mentioned paper, where it is a collection of random variables. Additionally, we clarify our choice of modeling the encoder-decoder attention map as a latent variable, which results in the stochastic nature of loss function.
>
> In the results section, we have added two extra experiments. First, we compare our method to the transformer model. To keep the evaluation scheme consistent for transformer baseline, we mix the generated samples with a set of clean/ground-truth samples and some noisy/distorted samples to eliminate listeners' bias as much as possible. Second, we have added a new subsection which evaluates the rhythm similarity between the target ground-truth and modified samples from the test set. Finally, we agree with the reviewer that duration modification alone cannot modify the underlying speaker and emotional information.

---

### Decision · Program_Chairs · 2023-06-14

**Decision:**

Accept

**Comment:**

SSW2003 received 45 papers. The acceptance rate is 82%. We are pleased to inform you that your paper has been accepted by the SSW2023 Program Committee. Please read the reviews carefully and submit your camera-ready paper by June 28th. Most of reviewers performed a detailed review. Please answer to their questions and take into account their comments.
Since your paper received a score below 5/9 that is strongly argued by the reviewers, note that the Program Committee will check if your manuscript has been significantly changed to specifically consider their remarks. Note that camera-ready papers are credited of one extra page to allow authors to consider reviewers’ suggestions. So max 7 pages in total including figures & refs.
The deadline for submitting the revised version (with full non anonymized authors and refs!) is 28th June.